# Two *Laminaria japonica* Fermentation Broths Alleviate Oxidative Stress and Inflammatory Response Caused by UVB Damage: Photoprotective and Reparative Effects

**DOI:** 10.3390/md20100650

**Published:** 2022-10-20

**Authors:** Qianru Sun, Jiaxuan Fang, Ziwen Wang, Zixin Song, Jiman Geng, Dongdong Wang, Changtao Wang, Meng Li

**Affiliations:** 1College of Chemistry and Materials Engineering, Beijing Technology & Business University, 11 Fucheng Road, Haidian District, Beijing 100048, China; 2Beijing Key Laboratory of Plant Resource Research and Development, Beijing Technology and Business University, 11 Fucheng Road, Beijing 100048, China

**Keywords:** photodamage, antioxidation, inflammation, *Laminaria japonica*, fermentation technology

## Abstract

UVB radiation can induce oxidative stress and inflammatory response in human epidermal cells. We establish a UVB-induced damage model of human immortalized epidermal keratinocytes (HaCaT) to explore the protective and reparative effects of *Laminaria japonica* on UVB-damaged epidermal inflammation after fermentation by white *Ganoderma lucidum* (*Curtis*) *P. Karst* and *Saccharomyces cerevisiae*. Compared with unfermented *Laminaria japonica*, fermented *Laminaria japonica* possesses stronger in vitro free radical scavenging ability. *Laminaria japonica* white *Ganoderma lucidum* fermentation broth (LJ-G) and *Laminaria japonica* rice wine yeast fermentation broth (LJ-Y) can more effectively remove excess reactive oxygen species (ROS) in cells and increase the content of the intracellular antioxidant enzymes heme oxygenase-1 (HO-1) and NAD(P)H quinone oxidoreductase 1 (NQO-1). In addition, fermented *Laminaria japonica* effectively reduces the content of pro-inflammatory factors ILs, TNF-α and MMP-9 secreted by cells. The molecular research results show that fermented *Laminaria japonica* activates the Nrf2 signaling pathway, increases the synthesis of antioxidant enzymes, inhibits the gene expression levels of pro-inflammatory factors, and alleviates cellular oxidative stress and inflammatory response caused by UVB radiation. Based on the above results, we conclude that fermented *Laminaria japonica* has stronger antioxidant and anti-inflammatory activity than unfermented *Laminaria japonica*, possesses good safety, and can be developed and used as a functional inflammation reliever. Fermented *Laminaria japonica* polysaccharide has a more slender morphological structure and more rockulose, with better moisturizing and rheological properties.

## 1. Introduction

As the largest organ of the human body, the skin is in direct contact with the external environment and performs such functions as protection, excretion and body temperature regulation [1]. The epidermis is responsible for resisting toxic substances in the external environment; it can withstand mechanical friction and microbial invasion, and effectively prevent the loss of skin moisture, ions and metabolites [2]. Healthy skin has the ability to renew itself, but when damaged by external stimuli and unable to repair and renew itself in time, it can become sensitive and inflamed, and even develop pathological reactions such as atopic dermatitis and psoriasis [3]. Exposure to UV radiation, environmental changes and unhealthy cleaning practices are all contributing factors, with UV radiation being among the most significant.

Ultraviolet radiation is divided into three categories according to the length of the wavelength band: UVA (315–400 nm), UVB (280–315 nm) and UVC (100–280 nm) [4]. UVB can reach the epidermis of the skin. Excessive exposure to UVB can cause skin erythema, sunburn, and damage to intracellular macromolecules [2]. Therefore, UVB is considered to have more serious damaging effects than UVA [5]. The UVB irradiation of the skin can cause an increase in the content of reactive oxygen species (ROS) in skin cells and trigger oxidative stress response [6]. Oxidative stress can cause many skin problems, such as inflammation and barrier damage [7]. Impaired skin barrier in atopic dermatitis is strongly associated with the secretion of cytokines (inflammatory factors) by keratinocytes. Inflammatory factors can inhibit the synthesis level of filaggrin, thereby affecting the process of stratum corneum differentiation. These changes lead to increased permeability to external antigens and the gradual breakdown of the skin barrier [8]. As the hole in the ozone layer enlarges, human exposure to UV radiation increases, as well as the probability of epidermal oxidative stress, inflammation and deep skin damage caused by UVB. Thus, there is an urgent need to find effective ways to alleviate skin oxidation and inflammation caused by photodamage.

NF-E2-related factor 2 (Nrf2) is a member of the CnC subfamily of leucine-pulled transcription factors in the basic region. Under normal conditions, Nrf2 binds to the protein element Kelch-1ike ECH-associated protein l (Keap-1), which is in an inactive state [9]. When stimulated or damaged, intracellular ROS induces a conformational change in Keap-1 protein to promote the dissociation and termination of the ubiquitination of Nrf2. The released Nrf2 is catalyzed by the small Maf protein and translocates into the nucleus, where it activates the transcriptional activity of a further series of genes [10].

As evidenced by existing studies, Nrf2 is an anti-inflammatory pathway. Inflammation typically occurs in Nrf2-deficient chemically-induced pathologies, and Nrf2-knockout mice have a significantly increased tendency to develop multi-tissue inflammatory lesions compared with normal mice [11]. The alleviation of inflammation by Nrf2 is associated with the inhibition of pro-inflammatory cytokine production, but the molecular mechanisms underlying the interaction between the two remain largely unclear [12]. The plant-derived anti-inflammatory agent 3-hydroxyanthranilic acid (HA) can promote Nrf2 nuclear translocation and induce heme oxygenase-1(HO-1) expression, suggesting that the anti-inflammatory effect of Nrf2 is partially dependent on redox dynamic regulation [13]. Studies have shown that synthetic triterpenoids can induce the increased expression and activity of NAD(P)H quinone oxidoreductase 1(NQO-1) in mice to reduce nitric oxide synthase (iNOS) levels, and this regulatory process is strictly dependent on Nrf2 [14].

*Laminaria japonica*, also known as brown seaweed, is common in East Asian countries such as China, Japan and South Korea as a traditional Chinese medicinal and edible algae [15]. It is rich in active substances, such as fucoidan, fucoxanthin, dietary fiber, and carotenoids, and possesses significant antioxidant [15], anti-inflammatory [16], anti-aging and antibacterial properties [17]. Studies have confirmed that *Laminaria japonica* specifically inhibits the nuclear translocation of the STAT1 gene through the MAPK pathway, reduces skin inflammation and the expression level of cellular inflammatory chemokines, and inhibits the loss of skin moisture in mice caused by dinitrochlorobenzene (NC/Nga) sensitization [18]. Fucoidan extracted from *Laminaria japonica* has been confirmed to reduce the content of tumor necrosis factor α (TNF-α) in THP-1 cells caused by lipopolysaccharide (LPS), and it can be reduced by 50% at 50 μg/mL [19]. Research on the anti-inflammatory activity of *Laminaria japonica* is emerging, but it remains blank in terms of UVB radiation.

Fermentation technology is a kind of natural plant active substance extraction technology that is currently mature and can convert macromolecular substances into small molecular active substances to achieve detoxification and efficiency enhancement [18]. Polysaccharides extracted from *Lycium barbarum* after fermentation by *Saccharomyces*
*cerevisiae* can significantly delay the aging process of nematodes, and possess stronger permeability [20]. Positive results have already been seen in terms of the antioxidant and disease treatment of *Laminaria japonica* treated with fermentation technology: extracts of *Laminaria japonica* fermented by *Aspergillus oryzae* have shown strong free radical scavenging effects, and some have even exceeded the same mass concentration of vitamin C [21]. The same results were obtained after the mixed fermentation of *Laminaria japonica* with two lactic acid bacteria. Fermentation treatment can not only effectively improve the extraction rate of the active substances in *Laminaria japonica*, but also promote stronger antioxidant and hypoglycemic activity [22].

In this study, traditional Chinese medicinal white *Ganoderma lucidum* and rice wine yeast are used to ferment *Laminaria japonica* in order to investigate whether the resulting fermentation broth has protective and reparative effects on skin photo-inflammation caused by UVB radiation, explore whether microbial fermentation can effectively improve the utilization rate of *Laminaria japonica* active substances and exert stronger anti-inflammatory effects, study the molecular mechanism of the effects of *Laminaria japonica* fermentation broth and provide a scientific basis for its application as an anti-inflammatory raw material.

## 2. Results

### 2.1. Changes in Active Substance Content

Laminaria japonica is rich in polysaccharides and polyphenols, which are the primary substances with antioxidant and anti-inflammatory effects [19,23].

The data showed that compared to Laminaria japonica water extract (LJ-W), the total sugar, polysaccharide, phenol and protein contents in the fermentation broth significantly increased (Table 1). The contents of total sugars, polysaccharides and proteins in Laminaria japonica white Ganoderma lucidum fermentation broth (LJ-G) and Laminaria japonica rice wine yeast fermentation broth (LJ-Y) showed an increase of 2–3 times. Although the content of total phenols in the three samples was not very high, the content after fermentation also increased by one. These experimental results confirm that the content of Laminaria japonica bioactive substances was significantly different before and after fermentation treatment. This suggests that fermentation can effectively improve the extraction rate of Laminaria japonica bioactive substances.

### 2.2. In Vitro Antioxidant Activity

Four types of free radical scavenging activity of *Laminaria japonica* water extract and fermentation broth were determined to evaluate the difference in antioxidant activity before and after fermentation (Figure 1A–D).

The scavenging data of the DPPH free radical test showed that the scavenging ability of the sample was weak in the concentration range of 0.25–4 mg/mL, and the scavenging effects of LJ-G and LJ-Y were worse than those of LJ-W. The clearance capacity of LJ-G was significantly improved after 4 mg/mL, and the clearance rate reached 76.75% at 8 mg/mL, which was significantly higher than that of LJ-W. The clearance rate of LJ-Y at 4–8 mg/mL was slightly improved, but there was no significant difference in the clearance between the two at the same concentration. The scavenging effects of samples on hydroxyl radicals were positively correlated with the mass concentration. The clearance of LJ-G was consistently better than that of LJ-W, reaching 86.63% and 87.09% at 4 and 8 mg/mL, respectively. The scavenging ability of LJ-Y was slightly weaker than that of LJ-G. It is worth noting that LJ-W showed the strongest scavenging ability on hydroxyl radicals at 8 mg/mL (63.51%). The same clearance was achieved by LJ-G and LJ-Y at 1 mg/mL, and the clearance rate of LJ-G was consistently higher than that of LJ-Y.

The test results of total antioxidant capacity determination (ABTS method) showed that both LJ-G and LJ-Y had significantly (*p* < 0.001) higher ABTS+ scavenging power than LJ-W, with 2.44 and 2.81 times that of LJ-W, respectively, and LJ-Y had the best scavenging ability, reaching 1.14 mM Trolox equivalent. The test results of total antioxidant capacity determination (FRAP method) showed that compared with LJ-W, the reducing power of LJ-G and LJ-Y were significantly improved (*p* < 0.001), and the reducing ability of the two on Fe^2+^ reached 1.55 and 1.28 times that of LJ-W respectively. Unlike the total antioxidant capacity (ABTS method), LJ-G showed a stronger reduction of Fe^2+^ than LJ-Y.

Based on the above in vitro antioxidant test data, we can conclude that LJ-W, LJ-G and LJ-Y all have certain in vitro antioxidant activity, while those of LJ-G and LJ-Y are significantly stronger than that of LJ-W. The scavenging activity of LJ-G on DPPH free radicals, hydroxyl radicals, and Fe^2+^ were the strongest among the three samples.

### 2.3. Effects of LJ-W, LJ-G, LJ-Y on the Viability of HaCaT Cells

The protective and reparative effects of LJ-W, LJ-G and LJ-Y on human immortalized epidermal keratinocytes (HaCaT) cells damaged by UVB are shown in Figure 2. UVB of 10 mJ/cm^2^ was selected as the experimental dose (Appendix A). In the protection group (Figure 2A), the cell viability of the three groups of samples was significantly improved compared with the model group. Cell viability increased gradually in the LJ-W group, reaching its highest level (114.17%) at 4 mg/mL. Cell viability in the LJ-G group was always greater than 100%, reaching its highest level (121.47%) at 2 mg/mL. Cell viability in the LJ-Y group was higher than that of the model group at a concentration of 0.125–2 mg/mL, and gradually increased with the concentration. At 4 mg/mL, the survival rate of HaCaT cells decreased to 73.70%, which was lower than that of the model group. The results of the repair group test (Figure 2B) showed that the cell survival rate after UVB irradiation was 72.39% (model group). The three samples with different concentrations showed similar trends in the repair of UVB-damaged HaCaT cells. With the increase in mass concentration, the cell viability of all three groups gradually increased, reaching a peak at 0.5 mg/mL, then gradually decreasing (0.5–2 mg/mL), but still remaining higher than that of the model group. At 4 mg/mL, the cell viability of the LJ-G group was still higher than that of the model group, but those of LJ-W and LJ-Y were lower.

The above experimental data shows that *Laminaria japonica* fermentation broth can effectively alleviate the damage and death of HaCaT cells caused by UVB, and promote the proliferation of epidermal cells, but high concentrations of LJ-Y are slightly toxic to cells. In the UVB protection group, the cell survival rate of *Laminaria japonica* fermentation broth reached its highest levels at 2 mg/mL, namely, 121.47% (LJ-G) and 106.08% (LJ-Y). The fermentation broth of the UVB repair group reached maximum cell viability at 0.5 mg/mL, namely, 95.63% (LJ-G) and 88.89% (LJ-Y). Therefore, we chose 2 mg/mL and 0.5 mg/mL as the sample concentrations for subsequent protection and repair experiments. For proper expression, the concentration unit of subsequent test samples was modified to “μg/mL”.

### 2.4. Effects of LJ-W, LJ-G and LJ-Y on ROS Inhibition and Scavenging

Excessive oxygen free radicals (ROS) will cross the cell membrane and react with most biomolecules (DNA, proteins, lipids, etc.), causing oxidative stress in the body and excessive levels of intracellular inflammatory factors, and eventually leading to the appearance of inflammatory skin problems [24]. Removing excess ROS is the guarantee for maintaining the normal functioning of the skin cells and body.

The effects of UVB damage and the protective and reparative effects of the samples on ROS content in HaCaT cells are shown in Figure 3. After UVB irradiation, the intracellular ROS content was significantly increased in the model group and significantly decreased in the protection and repair groups (*p* < 0.001). As shown by the intracellular ROS assay fluorescence values (Figure 3B), in the damage protection group, LJ-W, LJ-G and LJ-Y significantly reduced intracellular ROS contents, while the ROS contents of the LJ-G and LJ-Y groups were lower than those of the LJ-W and control groups. In the damage repair group, the intracellular ROS contents of the LJ-W and LJ-Y groups were lower than that of the control group. The scavenging effect of LJ-G was weaker than those of LJ-Y, but still significant (*p* < 0.001).

The above experimental data indicates that the water extract and fermentation broth of *Laminaria japonica* can effectively remove excess ROS in damaged HaCaT cells; the scavenging effects of *Laminaria japonica* fermentation broth are better than that of *Laminaria japonica* water extract; and LJ-G and LJ-Y in the protection group and LJ-Y in the repair group can reduce ROS in damaged cells to normal levels. This reveals that LJ-Y possesses enhanced physiological activity in ROS scavenging.

### 2.5. Effects of LJ-W, LJ-G, LJ-Y on Antioxidant Capacity

Heme oxygenase-1 (HO-1) is an antioxidant cytoprotective enzyme that plays a key role in oxidative stress and inflammation [25]. NQO-1 belongs to the DMES in the Phase II detoxification stage of cells and can protect the cytoplasmic membrane from ROS damage [14].

Figure 4 shows the effects of LJ-W, LJ-G and LJY on HO-1 and NQO-1 contents in cells. After UVB irradiation, compared with the blank group, the contents of HO-1 and NQO-1 in HaCaT cells were significantly decreased.

In the protection group, the effects of the three samples on HO-1 enzyme content in HaCaT cells were similar. However, after LJ-G treatment, the expression of HO-1 mRNA in cells was significantly upregulated, which was higher than LJ-Y. The LJ-G and LJ-Y groups significantly affected NQO-1 in HaCaT cells, showing excellent protective effects. Especially in LJ-G-treated HaCaT cells, the intracellular NQO-1 enzyme content was significantly increased, and its mRNA expression was four times that of the UVB-damaged model group. In the repair group, the range of antioxidant enzymes in the LJ-W group increased slightly but not significantly. LJ-G and LJ-Y could dramatically increase the content and transcriptional activity of HO-1 and NQO-1, and LJ-G was stronger.

To sum up, after UVB irradiation, while the three samples showed certain protective and reparative effects on cell oxidative stress damage, LJ-G exhibited stronger regulatory activity than LJ-Y, which will affect the activity of intracellular antioxidant enzymes and detoxification enzymes.

### 2.6. Effects of LJ-W, LJ-G, LJ-Y on the Expression Levels of Inflammatory Factors

Keratinocytes can secrete inflammatory chemokines (interleukins, tumor necrosis factor, etc.) and cytokines to maintain skin homeostasis. Elevated ROS levels caused by the UVB irradiation of epidermal cells can trigger the excessive secretion of inflammatory chemokines, which in turn causes the infiltration of T cells and neutrophils into the epidermis, leading to inflammation and skin lesions [18]. The elevated secretion of pro-inflammatory factors (interleukins and tumor necrosis factor) marks the beginning of cellular inflammatory response [26].

MMPs are also mediators of the vicious cycle of inflammation which is responsible for degrading various components in the extracellular matrix [27]. Studies have shown that Nrf2-knockout skin ulcer wounds heal slowly, which is attributed to the high expression of MMP-9 in the cells [28]. After Nrf2-knockout, mice are irradiated with UVB, the activity of MMPs in the skin cells significantly increases, indicating that Nrf2 can play a photoprotective role by inhibiting MMPs [29].

*Laminaria japonica* water extract and fermentation broth have significant effects on inflammatory chemokines interleukin-1β (IL-1β), interleukin-8, IL-8 (IL-8), tumor necrosis factor-α (TNF-α) and matrix metalloproteinase-9 (MMP-9). The effects of expression level are shown in Figure 5. After UVB irradiation, the contents and gene transcription levels of IL-1β, IL-8, TNF-α and MMP-9 in the protection and repair groups were significantly increased. In the protection group (Figure 5A), the levels of TNF-α and MMP-9 in the LJ-W group were significantly lower than those in the model group, but there was no significant difference in the levels of IL-1β and IL-8. In the LJ-G and LJ-Y groups, the contents and transcription levels of the four pro-inflammatory factors were significantly lower than those in the model group and LJ-W group. In the repair group (Figure 5B), LJ-W was able to reduce IL-1β and MMP-9 content but had no significant effect on the other inflammatory factors. The contents of the above pro-inflammatory factors were significantly decreased in the LJ-G and LJ-Y groups, and the contents of IL-1β and TNF-α in the LJ-G group were even lower than those in the LJ-Y and control group.

The above results indicate that UVB irradiation can induce the excessive secretion of intracellular inflammatory chemokines, leading to cellular inflammation. *Laminaria japonica* has a certain anti-inflammatory activity which is amplified by fermentation technology. It is worth noting that the inhibitory effects of the two *Laminaria japonica* fermentation broths on inflammatory factors were different in terms of protection and repair. In the protection group, LJ-Y showed a better ability to inhibit inflammation, while LJ-G showed a better ability in the repair group. Compared to LJ-Y, LJ-G could reduce the levels of inflammatory factors even more than undamaged cells. In conclusion, both the water extract and fermentation broth of *Laminaria japonica* have anti-inflammatory activity, and that of the fermentation broth is better than that of the water extract.

### 2.7. Effects of LJ-W, LJ-G, LJ-Y on Keratinocyte Differentiation and Skin Barrier Function

The stratum corneum of the skin is rich in functional proteins and active enzymes, such as aquaporin 3 (AQP3), filaggrin (FLG), kallikrein-7 (KLK-7) and Caspase-14. AQP3 is located on the cell membrane and is responsible for tightly bonding skin keratin [30]. FLG forms a tight physical barrier in the epidermis to prevent water loss and the invasion of external irritants [31]. KLK-7 is responsible for severing the desmosomes that connect cells, causing skin cell dysfunction and accelerating cell shedding [30]. Caspase-14 is active in the damaged epidermis and can hydrolyze FLG into the intercellular natural moisturizing factor (NMF) to maintain the moisturizing function of the skin [32].

The levels of AQP3 and FLG in the cells were significantly decreased after UVB irradiation, while the levels of KLK-7 and Caspase-14 were significantly increased (Figure 6). This indicates that UVB irradiation accelerates intracellular water loss and leads to damaged epidermal cell shedding.

After the effect of the samples, the content of water retention protein in the cells was significantly increased and the enzymatic activities of KLK-7 and Caspase-14 were significantly decreased. The protection and repair groups showed similar results. Interestingly, the barrier repair activity of LJ-G was higher than that of LJ-Y in the repair group, whereas LJ-Y was better in the protection group.

In summary, it is likely that LJ-W, LJ-G and LJ-Y can maintain the normal connections between cells. On the premise of maintaining the stability of the extracellular matrix and NMF contents, the reinforcement of filaggrin in the skin barrier is enhanced to ensure the normal function of the stratum corneum barrier, thereby effectively protecting the epidermis from UVB radiation damage.

### 2.8. Effects of LJ-W, LJ-G, LJ-Y on Gene Expression of Nrf2 Signaling Pathway

The Nrf2 signaling pathway can regulate cellular inflammatory response [27]. The Nrf2 gene bound to the Keap-1 element is silent. At this time, the transcription of downstream genes p38, JNK1 and AP-1 is active, and can promote the expression of IL-1β, TNF-α and MMP-9. After separation from Keap-1, Nrf2 transcriptional activity is activated and translocated into the nucleus, where it regulates the downstream genes to synthesize antioxidant enzymes and inhibit the expression of pro-inflammatory factors.

We measured the transcriptional activity of the Nrf2 signaling pathway-related node genes Keap-1, p38, JNK1 and AP-1 in damaged cells before and after treatment by the samples (Figure 7). The data showed that the transcriptional activity of Keap-1 in UVB-irradiated cells was increased, the transcriptional level of the Nrf2 pathway was decreased and the transcriptional levels of its downstream genes p38, JNK1 and AP-1 were significantly increased (*p* < 0.001). After the samples were applied, the transcriptional activity of Keap-1 decreased, the transcriptional level of Nrf2 increased and the transcriptional activity of p38, JNK1 and AP-1 decreased. In the protection group (Figure 7A), the transcriptional activity of Nrf2 was significantly higher in the LJ-G and LJ-Y groups than in the LJ-W group (*p* < 0.05). In addition, LJ-G and LJ-Y had stronger inhibitory effects on p38, JNK1 and AP-1 than LJ-W. The repair group (Figure 8B) data also showed the same trend as the protection group. Notably, in the protection and repair group, the regulatory activity of LJ-G was higher than that of LJ-Y for most node genes.

The above experimental results showed that UVB irradiation could inhibit the translocation of the Nrf2 signal into the nucleus in HaCaT cells, reduce the body’s antioxidant and detoxification process, and prompt cells to produce a large number of inflammatory chemokines, causing an inflammatory response. *Laminaria japonica* improved the antioxidant effects of cells by increasing the transcription level of Nrf2 and promoting its nuclear translocation. *Laminaria japonica* also reduced the transcriptional activity of AP-1, which, in turn, suppressed the high expression levels of IL-1β, TNF-α and MMP-9 induced by UVB irradiation.

### 2.9. LJ-W, LJ-G, LJ-Y Safety Evaluation

The chick chorioallantoic membrane eye irritation test (CAM) and erythrocyte hemolysis test can determine whether a tested substance is an irritant through the detection of vascular bleeding and hemolysis [33]. Thus, CAM and erythrocyte hemolysis tests were performed on LJ-W, LJ-G and LJ-Y to evaluate their safety (Figure 8A–D). Sodium dodecyl sulfate (SDS) was used as a positive control. The results showed that samples with different mass concentrations had little effect on the hemolysis of rabbit erythrocytes (Figure 8A–C). At the same concentration, the hemolysis degrees of LJ-G and LJ-Y were lower than that of LJ-W. 0.9% NaCl and 0.1 mol/L NaOH were used in control group and model group controls, respectively (Table 2). After the NaOH (model group) was used for 3 min, the blood vessels on the membrane showed obvious hemorrhage, vascular coagulation and vascular dissolution caused by severe stimulation, while the sample group and negative control group did not show the above phenomena. According to these results, we believe that *Laminaria japonica* fermented freeze-dried powder is non-irritating to the eyes, has high safety and can be added to skin care products as a functional raw material.

### 2.10. Physical Properties of LJ-Wp, LJ-Gp and LJ-Yp

The FT-IR pattern of *Laminaria japonica* is shown in Figure 9A. The absorption peaks of LJ-Wp (a), LJ-Gp (b) and LJ-Yp (c) at 3350 cm^−1^ were caused by the C-H triple bond (alkynyl) and hydroxyl stretching vibration. It is speculated that there may be a multi-molecular association structure. The absorption peaks at 3033 and 2923 cm^−1^ are caused by the stretching vibration of the carbon-hydrogen single bond. The sharp absorption peak at 1612 cm^−1^ was caused by carbonyl stretching and benzene ring skeleton vibration. The absorption peaks at 1408, 1244 and 1053 cm^−1^ are caused by the stretching of carbon single bond and carbon-hydrogen bond, and the absorption peak at 1408 cm^−1^ suggests the existence of halogen. The smaller sharp absorption peak at 826 cm^−1^ was caused by the disubstituted benzene ring, which was confirmed by the absorption peak at 1612 cm^−1^. Comparing the FI-IR results of LJ-Wp, LJ-Gp and LJ-Yp, we found that LJ-Gp has no obvious absorption peak at 826 cm^−1^, and the absorption peak area at 1612 cm^−1^ is correspondingly reduced. It is speculated that white *Ganoderma lucidum* may have changed the original benzene ring structure of *Laminaria japonica* during the fermentation process.

The relative molecular weights of LJ-Wp, LJ-Gp and LJ-Yp were determined using gel permeation chromatography (GPC). The elution curve is shown in Figure 9B. The measured weight average molecular weights (Mw) of LJ-Wp (a), LJ-Gp (b) and LJ-Yp (c) were 16.1232 × 10^4^, 18.1159 × 10^4^ and 39.6117 × 10^4^ kDa, respectively (Table 3). The molecular weight of LJ-Gp and LJ-Yp was higher than that of LJ-Wp, indicating that fermentation technology can improve the *Laminaria japonica* wall-breaking rate and promote the release of macromolecular polysaccharide active substances.

We used scanning electron microscopy (SEM) to observe the structure and morphology of *Laminaria japonica*. Figure 9C showed the topographic features under different observation accuracies. LJ-Wp (a), LJ-Gp (b) and LJ-Yp (c) all showed irregular loose sheet-like or filamentous structures. Compared with LJ-Wp, LJ-Gp has more filamentous structure, and LJ-Yp structure is smaller and less sheet-like structure. The microstructure of fermented *Laminaria japonica* polysaccharide showed loose pores and smaller structural morphology. It is speculated that LJ-Gp and LJ-Yp have better rheological properties and absorption.

### 2.11. Monosaccharide Components of LJ-Wp, LJ-Gp and LJ-Yp

Table 4 and Figure 10 shows the monosaccharide composition of LJ-Wp, LJ-Gp and LJ-Yp. The highest content was found in rockulose, followed by galactose, glucuronide and mannose. The contents of fucose in LJ-Wp, LJ-Gp and LJ-Yp were 45.467, 39.620 and 44.675%, and the contents of galactose were 22.094, 21.020 and 26.090%. The content of glucuronic acid and mannose is about 10–13%. Fucoidan is widely used as a moisturizer in skin care products, and mannose can regulate the micro-ecological balance of the skin [34,35]. It is presumed that *Laminaria japonica* polysaccharide has better moisturizing effect and skin conditioning activity.

## 3. Discussion

Inflammation is associated with many skin problems and physical diseases, and sun exposure is one factor that causes skin inflammation. UVB in sunlight can reach the epidermis of the skin, stimulate cells to produce an inflammatory response and cause other skin problems. Fucoidan extracted from *Laminaria japonica* exhibits positive in vitro anti-inflammatory activity, confirming that *Laminaria japonica* has significant anti-inflammatory effects [19], but our understanding of the role of *Laminaria japonica* in combating photo-inflammation is unclear.

The generation of photo-inflammation is complex and multidimensional. Sunlight directly stimulates skin epidermal cells to secrete pro-inflammatory factors [36]. Epidermal cells exposed to sunlight undergo oxidative stress, and excess ROS stimulates the cells to express more inflammatory chemokines [37]. The production of inflammation can lead to the destruction of the tight junction function of the epidermis, and damage to the barrier will increase the permeability of the skin to external stimuli, leading to a vicious circle of inflammation accumulation [38]. We established a UVB radiation damage system and used HaCaT cells to simulate human epidermal cells under sunlight in order to explore the protective and reparative effects of *Laminaria japonica* on epidermal inflammation at the in vitro, cellular and molecular levels. The results showed that the UVB irradiation of HaCaT cells caused changes in the contents of various substances. UVB irradiation produces a large amount of ROS in cells, which accelerates the consumption of antioxidant enzymes stored in cells and reduces their original antioxidant capacity. In addition, UVB radiation aggravates the expression of inflammatory factors, including interleukin, tumor necrosis factor, and MMPs in cells, which directly leads to the generation of epidermal inflammation.

It has been confirmed that *Laminaria japonica* possesses excellent antioxidant and anti-inflammatory properties and is rich in a variety of biologically active substances. Studies have shown that macrocystis-derived fucoidan has strong anti-inflammatory effects at low concentrations, while *Laminaria japonica*-derived macromolecular fucoidan requires five times the dose to achieve the same level [19]. However, they have very similar anti-inflammatory activity at the same molecular weight. In addition, the tough cell walls of the seaweed are a factor that hinders active substance extraction, but fermentation technology can use enzymes produced by microorganisms to increase cell wall permeability and carry out the small molecular modification of the precipitated active substances [39]. Therefore, we chose to ferment *Laminaria japonica* in order to compare and explore the effects of fermentation on its anti-inflammatory activity.

In the study, we found that all *Laminaria japonica* samples showed good antioxidant and anti-inflammatory activity. Among them, the effects of LJ-G and LJ-Y were better than that of LJ-W. In terms of antioxidation, LJ-G and LJ-Y in the protection and repair groups could promote the expression of antioxidant enzymes HO-1 and NQO-1 in the body to different degrees, while LJ-G showed higher physiological activity. In terms of inflammation inhibition, we chose to measure the most typical pro-inflammatory factors (interleukin and tumor necrosis factor). The results showed that LJ-G and LJ-Y had significant inhibitory effects on the expression levels of inflammatory factors in HaCaT cells. Interestingly, in the protection group, LJ-Y had a better inhibitory effect on interleukin, while LJ-G had stronger inhibitory activity on TNF-α and MMP-9. In the repair group, the comprehensive anti-inflammatory activity of LJ-G was stronger. Whether the differences between LJ-G and LJ-Y in the protection and repair groups were related to different bacteria fermentation methods or caused by differences in the anti-inflammatory activity of *Laminaria japonica* itself are questions that remain to be explored in the future.

Nrf2 can establish a glutathione-mediated UVB detoxification mechanism in the epidermis, alleviate the toxic effects of UVB on epidermal cells and inhibit the apoptosis of mouse epidermal cells [40]. Malonic acid extracted from Korean pine can increase the levels of antioxidant enzymes HO-1 and SOD by regulating the Nrf2 signaling pathway, and increase the survival rate of HaCaT cells by inhibiting ROS, confirming that it can inhibit skin inflammation [41]. In addition, Nrf2 affects the downstream AP-1 gene, which in turn affects the expression of genes such as IL-1β, TNF-α and MMPs, and regulates the body’s inflammatory response process [27]. The results of this study show that *Laminaria japonica* can affect the expression levels of key node genes in the Nrf2 signaling pathway, promote its activation and nuclear translocation, inhibit the expression level of downstream genes, and exert its anti-inflammatory activity. Among the five node genes we explored, the LJ-G group had the most significant regulatory activity on Nrf2 and AP-1 in the protection and repair process, which are the target genes that directly regulate the inflammatory response. Combined with the research results at the cellular level, it can be considered that LJ-G has stronger anti-inflammatory activity.

In this study, fermentation technology was used to treat *Laminaria japonica*, and the damage effects of UVB on HaCaT cells and the protection and repair activity of *Laminaria japonica* on damaged cells were explored at the in vitro, cellular, and molecular levels. The results showed that after fermentation, the total sugar, polysaccharide, protein and total phenol contents in *Laminaria japonica* extracts increased significantly. This indicates that fermentation can effectively improve the extraction rate of its active substances. The in vitro antioxidant test results showed that LJ-G and LJ-Y had better in vitro oxygen-free radical scavenging activity than LJ-W, and that of LJ-G was better. In this study, a UVA-irradiated HaCaT cell inflammatory damage model was established, and a suitable concentration of *Laminaria japonica* extract was screened to investigate its anti-inflammatory activity at the cellular and molecular levels. The results showed that *Laminaria japonica* fermentation broth could activate the Nrf2 signaling pathway and promote the synthesis of antioxidant enzymes by promoting the dissociation of Nrf2 and Keap-1. In addition, *Laminaria japonica* fermentation broth regulated the transcriptional activity of Nrf2 downstream genes p38, JNK1, and AP-1, and inhibited the production of pro-inflammatory factors IL-1β, IL-8, TNF-α, and MMP-9, effectively alleviating the epidermal inflammatory response caused by UVB irradiation. The interaction of the high expression of pro-inflammatory factors and barrier disruption can lead to the aggravation of epidermal inflammation. The measurement results of skin barrier differentiation markers proved that *Laminaria japonica* fermentation broth alleviated the degree of damage to the skin barrier. This is supported by the fermentation broth’s inhibition of the decomposition of the extracellular matrix by MMP-9. In this study, the eye irritation risk of *Laminaria japonica* extract was assessed using the erythrocyte hemolysis and CAM tests. The results showed that *Laminaria japonica* extract is safe and can be used as an anti-inflammatory raw material for the development and research of products related to skin inflammation. In addition, the loose filamentous structure of fermented *Laminaria japonica* polysaccharide and the monosaccharide fraction with rockulose as the main component also implied that it had good moisturizing and absorption effects.

This study used fermentation technology to treat *Laminaria japonica* and confirmed that fermentation can improve the utilization of active substances and increase anti-inflammatory activity. We compared *Laminaria japonica* alcoholic extracts with anti-inflammatory activity and found that LJ-G and LJ-Y could exert better anti-inflammatory and skin barrier repair effects [16]. Compared with the alcoholic extraction process, the *Laminaria japonica* extracts obtained using the fermentation method have a high extraction rate, high safety and high eco-friendliness that traditional organic solvent extraction lacks. However, we did not quantify the landmark anti-inflammatory active substances in the fermentation broth. We did not consider the effects of the fermented strain’s own secretions on the physiological activity of *Laminaria japonica* fermentation broth. In the future, further physiological activity studies should be carried out on the anti-inflammatory active substances in the fermentation broth. In addition, does *Laminaria japonica* fermentation broth positively affect UVB-induced epidermal differentiation and defense mechanisms? Does *Laminaria japonica* fermentation relieve inflammatory skin diseases? In the future, more detailed and in-depth scientific research should be conducted on the photoprotection and repair mechanism of *Laminaria japonica* on the skin in order to further explore the biological properties of this abundant algae.

## 4. Materials and Methods

### 4.1. Preparation of Laminaria japonica Water Extract and Two Fermentation Broths

*Laminaria japonica* (Dalian, China) was crushed and passed through a 50-mesh sieve, dissolved in deionized water at a solid-liquid ratio of 1:40, extracted in a water bath at 80 °C for 5 h, and centrifuged at 4800 rpm for 30 min, and the supernatant was taken to obtain a *Laminaria japonica* water extract (named LJ-W) which was lyophilized through a membrane for use.

A single colony of rice wine yeast (Culture Collection Center of the Beijing Institute of Food and Brewing, Beijing, China) on YPD medium was placed in a 28 °C incubator for 48 h at 180 rpm to activate the strain. White *Ganoderma lucidum* (laboratory collection, deposit number: CGMCC No. 17789) was inoculated on a solid PDA medium and cultured at 28 °C for 12 d. The mycelium of white *Ganoderma lucidum* was picked and inoculated into a liquid PDA medium for 7 d to obtain mycelium balls.

Pretreatment was the same as that for LJ-W. *Laminaria japonica* powder was dissolved in deionized water and inserted into white *Ganoderma lucidum* mycelium balls and rice wine yeast at a material-to-liquid ratio of 1:20, then placed in an incubator and incubated at 180 rpm and 28 °C for 48 h. It was then centrifuged at 4800 rpm for 30 min, and the supernatant was taken to obtain white *Ganoderma lucidum Laminaria japonica* fermentation broth (LJ-G) and rice wine yeast *Laminaria japonica* fermentation broth (LJ-Y), which were lyophilized through a membrane for use.

### 4.2. Determination of Changes in Active Substance Content

Determination of total sugar content: a BC2710 Total Sugar Kit (Solarbio Biotechnology Co., Ltd., Beijing, China) was used to determine the total sugar content of LJ-W, LJ-G and LJ-Y, and the operation steps in the kit’s instructions were followed.

Determination of reducing sugar content: glucose standard solution was prepared for use, 1 mg/mL of standard solution at different concentrations were taken and 2 mg/mL of DNS reagent (Carnos Technology Co., Ltd., Wuhan, China) was added. After cooling in a boiling water bath for 2 min, it was diluted five times with deionized water, the OD540 was measured, and a reducing sugar standard curve was drawn. The above steps were repeated to determine the reducing sugar content of LJ-W, LJ-G and LJ-Y.

Determination of total phenolic content: 1 mL of the standard solution of pyrogallic acid at different concentrations was prepared, 1 mL of deionized water, 0.5 mL of 2-fold diluted folin-phenol, and 1.5 mL of Na_2_CO_3_ solution with a mass fraction of 26.7% were added, and it was diluted 2.5 times at room temperature. After reacting for 2 h, the OD760 was determined, and a total phenol standard curve was drawn.

Determination of protein content: a BCA Protein Concentration Assay Kit (Biorigin Biotechnology Co., Ltd., Beijing, China) was used to determine the protein content of LJ-W, LJ-G and LJ-Y, and the operation steps in the kit’s instructions were followed.

### 4.3. In Vitro Antioxidant Activity Assay

The experimental steps of DPPH and hydroxyl radical scavenging were implemented with reference to the literature [30].

Total antioxidant capacity determination: the total antioxidant capacity of the samples was determined using a Total Antioxidant Capacity Detection kit (ABTS and FRAP methods) (Beyotime Biotechnology Co., Ltd., Shanghai, China). According to the kit’s instructions, Trolox and FeSO_4_•7H_2_O standard solution was configured, a standard curve was drawn and the total antioxidant capacity and Fe^2+^ reducing the power of LJ-W, LJ-G and LJ-Y was measured.

### 4.4. Cell Culture, UVB Irradiation Model Establishment and CCK8

HaCaT cells were purchased from the Cell Resource Centre (Beijing Union Medical College, Beijing, China). Cells were grown in a DMEM medium (Gibco, San Francisco, CA, USA) supplemented with 2% penicillin-streptomycin sulfate (Thermo Fisher Scientific Co., Ltd., Shanghai, China) and 10% fetal bovine serum (Gibco, San Francisco, CA, USA) at 37 °C in a humidified environment with 5% CO_2_. HaCaT cells were inoculated in T75 culture plates (Corning, Corning, NY, USA) at an inoculation density of 1 × 10^5^/mL (the total volume of culture solution in the bottle is about 16 mL), adhered to the bottom of the plate and grew 24 h.

HaCaT cell viability was determined using the CCK8 colorimetric method. One hundred microliters of cell suspension (1 × 10^4^/well) was inoculated into each well of a 96-well plate and it was placed in a 37 °C CO_2_ incubator for 24 h. The supernatant was aspirated, 100 μL of PBS was added, different doses (0, 5, 10, 20, 30, 40 mJ/cm^2^) of UVB were irradiated by UV crosslinker SCIENTZ03-II (SCIENTZ Biotechnology Co., Ltd., Ningbo, China). The total UVB energy was set at 40 mJ/cm^2^ and a double layer of tinfoil was used to cover the 96-well plate, with 100 μL PBS added in the middle of each column of cells to reduce interference. The tinfoil position was adjusted during the irradiation process until the energy was depleted. PBS was aspirated and replaced by basal medium, 10 μL of CCK8 was added to each well and the OD450 was measured after incubation for 2 h. Then the cell viability was calculated, where the cell survival rate was calculated using the ratio of UVB damage group to control group multiplied by 100%.

The sample concentration selection procedure was the same as above. The difference is that the samples (100 μL/well) were added after UVB damage and continued to incubate for 12 h, control and model groups used equal amounts of basal medium instead of sample.

### 4.5. ROS Assay

To damage HaCaT cells, 10 mJ/cm^2^ UVB was used. References [42,43] performed ROS content determination on damaged HaCaT cells that cultured in a 6-well plate. A ROS Detection Kit was selected for the experiment, the intracellular ROS fluorescence content was observed under a fluorescence inverted microscope (CKX53, Olympus LS, Tokyo, Japan) and the fluorescence intensity was recorded. The treatment of the control group and model group was carried out simultaneously with that of the sample group. During the process of damage and sample treatment in the sample group, the control and model groups were given the same amount of serum-free DMEM for culture.

### 4.6. Determination of Inflammation and Glial Differentiation Factor Secretion by ELISA

HaCaT cells (30 × 10^4^/well) were cultured in a 6-well plate, the culture supernatant was collected and centrifuged at 4 °C and 6950 g for 10 min, and the supernatant was taken. ELISA kits (Wuhan Huamei Biological Engineering Co., Ltd., Wuhan, China) were used to detect the pro-inflammatory factors IL-1β, IL-8, TNF-α, MMP-9, AQP3, KLK-7, FLG, and Caspase-14.

### 4.7. qRT-PCR

HaCaT cells (30 × 10^4^/well) were cultured for 24 h in a 6-well plate in order to make its cell count about 100 × 10^4^/well, and the final volume was 2 mL per well. Total RNA extraction and reverse transcription were used by Trizol and First Stand cDNA. cDNA obtained by further reverse transcription was tested using Fast Super EvaGreen^®^ qPCR Master Mix and qRT-PCR (QuantStudio 3, Thermoscientific, Shanghai, China). The relevant primer sequences are shown in Table 5. The primer sequences are shown in Table 6. And the qRT-PCR reaction system is shown in Table 7.

### 4.8. Extraction of Polysaccharides from Laminaria japonica Fermentation Broth

The polysaccharide in Laminaria japonica fermentation broth was extracted by alcohol precipitation method. Dissolve the lyophilized powder prepared in 4.1 in deionized water, add three times the volume of absolute ethanol, and store at 4 °C overnight. Reconstitute the pellet with 100 mL water to the original volume and add three times the volume of Savage reagent to elute the protein. Using the alcohol precipitation method again, the precipitate was collected. The obtained precipitates were dissolved in ultrapure water and dialyzed (with 10 kDa MW cut-off) for 48 h, then freeze-dried to obtain crude polysaccharides, named “LJ-Wp, LJ-Gp, and LJ-Yp”.

### 4.9. FT-IR, GPC and SEM

LJ-Wp, LJ-Gp, and LJ-Yp were analyzed by infrared spectroscopy using the potassium bromide tablet method, and FI-IR was analyzed by Vertex 70 FT-infrared spectroscopy in transmission mode. The wavelength range is 400–4000 cm*^−^*^1^, and the resolution is 1 cm*^−^*^1^.

The molecular weights of LJ-Wp, LJ-Gp, and LJ-Yp were determined by GPC-LS-IR, and the morphological characteristics of them were determined using an FEI Nova Nano SEM 450 instrument. The specific experimental method refer to Zhang et al. [43].

### 4.10. Monosaccharide Composition Analysis

Precisely weighed reference standards such as mannose, rhamnose and glucose, dissolved in water and dilute to 50 μg each in 1 mL of liquid. Precisely weighed the sample into a 10 mL ampoule bottle, added 3.0 mL of 2 mol/L TFA, filled with nitrogen, seal the tube, and hydrolyzed it with acid at 120 °C for 4 h. Took out and added methanol nitrogen to dry TFA, added 3.0 mL of water to reconstitute, and calculate the concentration of 50 μg/mL.

Drew the mixed standard solution, 0.6 mol/L NaOH, 0.4 mol/LPMP-methanol, the volume ratio was 1:1:2, and reacted at 70 °C for 1 h. Cooled in cold water for 10 min; added 0.3 mol/L HCl to neutralize, then added 1 mL of chloroform, vortex for 1 min, centrifuge at 3000 r/min for 10 min, and extracted three times. The supernatant was used for HPLC. The sample derivatization process repeats the above steps, except that the mixed standard solution is replaced by the sample solution.

The HPLC measurement used Xtimate C18 4.6 × 200 mm 5 μm chromatographic column, the mobile phase was composed of 83% 0.05 M potassium dihydrogen phosphate solution and 13% acetonitrile, the flow rate was 1.0 mL/min, and the injection volume was 20 μL.

### 4.11. Data Analysis

SPSS software (SPSS, Version 17.0, IBM Inc., Armonk, NY, USA) was used for data analysis. Origin 2018 was used for data visualization. All experiments were performed in triplicate, and data is presented as mean ± standard deviation. T-test was used to determine the significance of differences between the groups (^NS^
*p* > 0.05, ^#,^* *p* < 0.05; ^##,^** *p* < 0.01, ^###,^*** *p* < 0.001). When *p* < 0.05, the difference is considered statistically significant.

## 5. Conclusions

*Laminaria japonica* has anti-inflammatory and skin barrier repair activity. We have successfully demonstrated that *Laminaria japonica* fermentation broth can effectively modulate UVB-induced epidermal inflammation and barrier damage. In this study, *Laminaria japonica* fermentation broths were shown for the first time to inhibit the production of inflammatory chemokines, which are the main triggers of inflammation. *Laminaria japonica* fermentation broth has also been shown to repair UVB-induced epidermal cell detachment and barrier breakage. The fermented *Laminaria japonica* polysaccharide had a more sparse and porous structure. Further studies will provide more insight into the activity of fermented polysaccharides from *Laminaria japonica* to test potential applications as a component against photo-inflammation.

## Figures and Tables

**Figure 1 marinedrugs-20-00650-f001:**
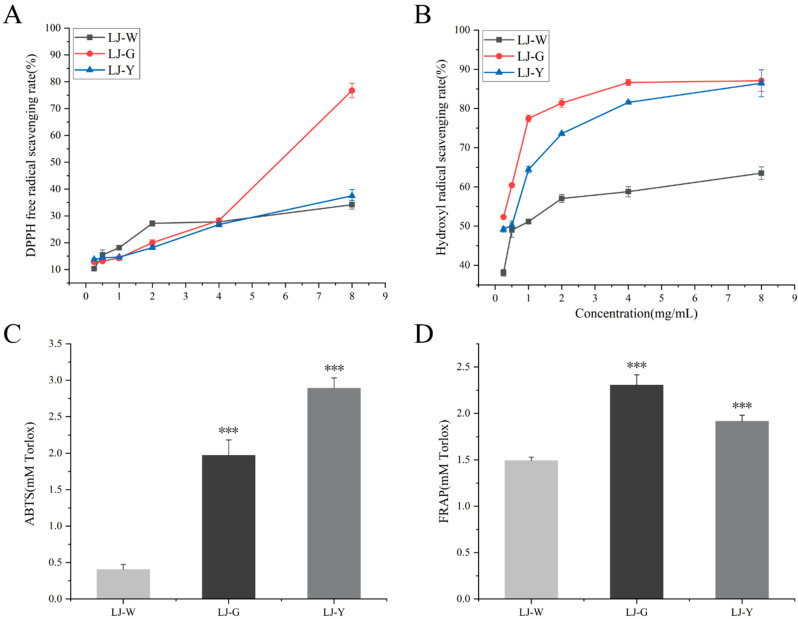
Antioxidant capacity of LJ-W, LJ-G and LJ-Y determined in vitro. (**A**): DPPH free radical scavenging ability; (**B**): hydroxyl radical scavenging ability; (**C**): total antioxidant capacity (ABTS method); (**D**): total antioxidant capacity (FRAP method). *** *p* < 0.001, significantly different compared to the LJ-W group.

**Figure 2 marinedrugs-20-00650-f002:**
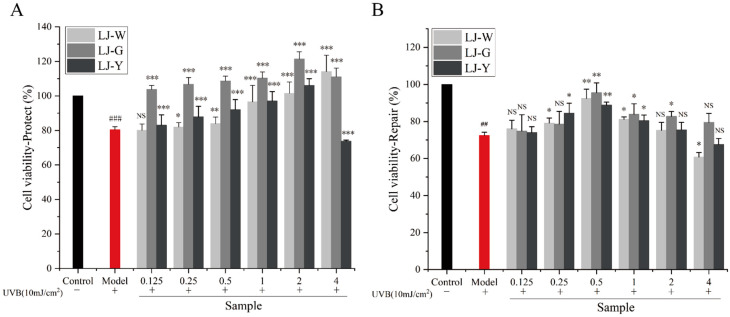
Effects of LJ-W, LJ-G and LJ-Y on the survival state of UVB-damaged HaCaT cells. The model group is the cell survival after UVB damage (red columns). (**A**): Protective effects of LJ-W, LJ-G and LJ-Y before UVB damage; (**B**): Reparative effects of LJ-W, LJ-G and LJ-Y after UVB damage. ^NS^ *p* > 0.05, * *p* < 0.05, ** *p* < 0.01, *** *p* < 0.001 compared with the model group, ^##^ *p* < 0.01, ^###^ *p* < 0.001 compared with the control group.

**Figure 3 marinedrugs-20-00650-f003:**
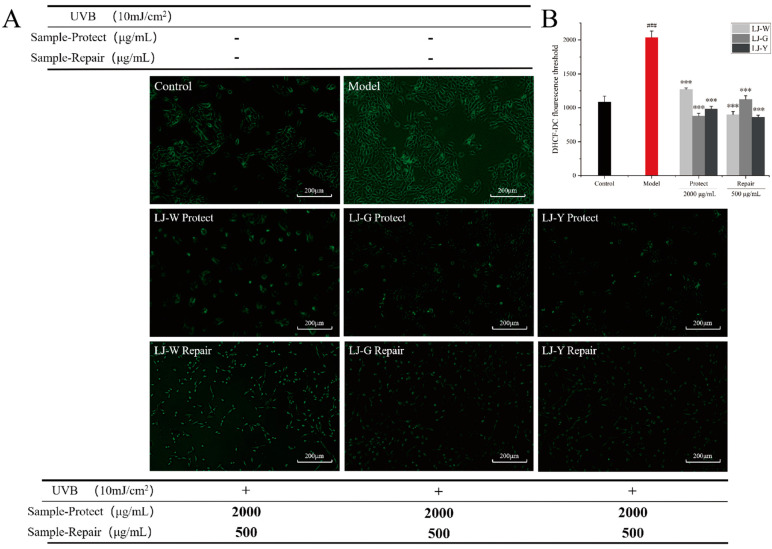
Effects of UVB, LJ-W, LJ-G and LJ-Y on intracellular ROS content. The model group is the intracellular ROS fluorescence values after UVB damage. (**A**): Cell ROS fluorescence intensity; (**B**): Cell fluorescence intensity value. *** *p* < 0.001 compared with the model group, ^###^ *p* < 0.001 compared with the control group.

**Figure 4 marinedrugs-20-00650-f004:**
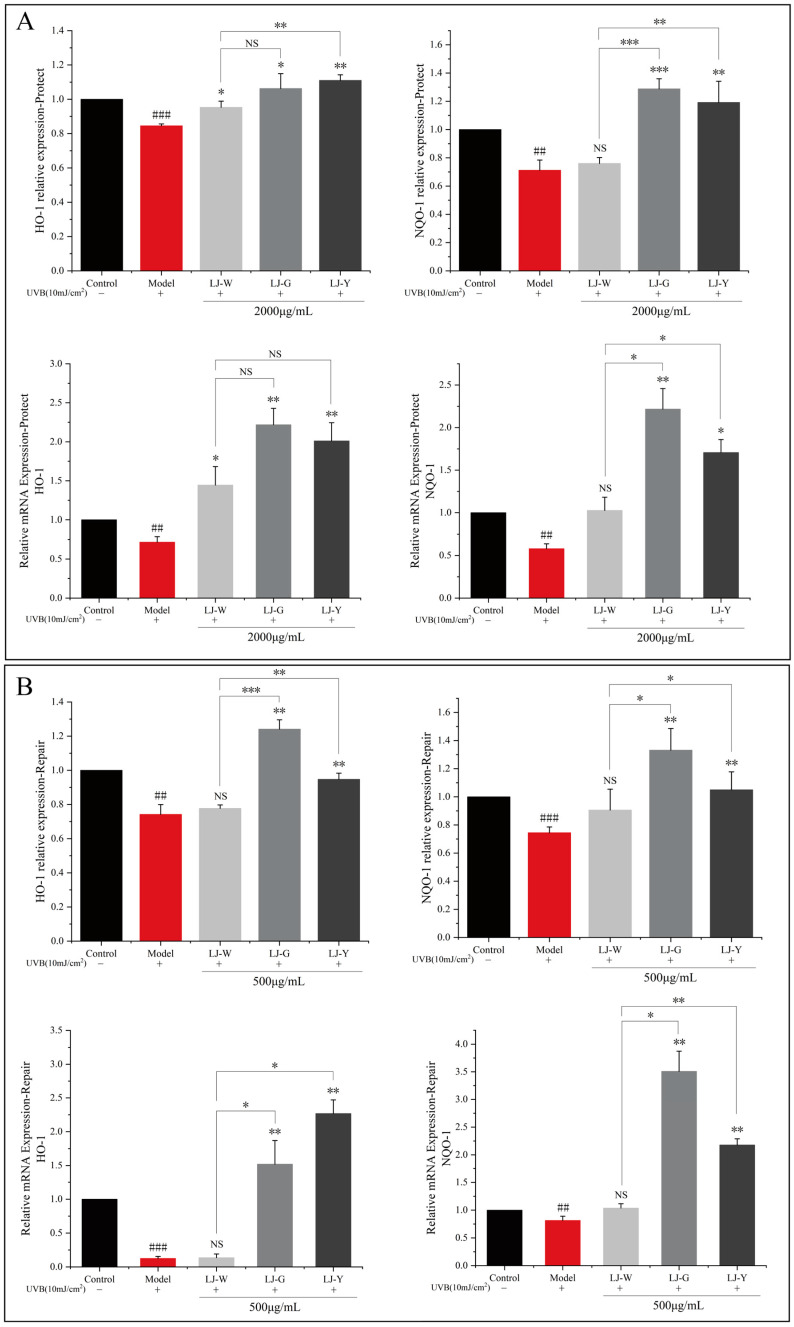
Effects of LJ-W, LJ-G and LJ-Y on the activity of antioxidant and detoxification enzymes in HaCaT cells. The model group was the relative expression content of intracellular inflammatory factors after UVB damage. (**A**): Differences in expression of HO-1, NQO-1 and related mRNAs in UVB-damaged protection group; (**B**): Differences in expression of HO-1, NQO-1 and related mRNAs in UVB-damaged repair group. ^NS^ *p* > 0.05, * *p* < 0.05, ** *p* < 0.01, *** *p* < 0.001 compared with the model and LJ-W groups, ^##^ *p* < 0.01, ^###^ *p* < 0.001 compared with the control group.

**Figure 5 marinedrugs-20-00650-f005:**
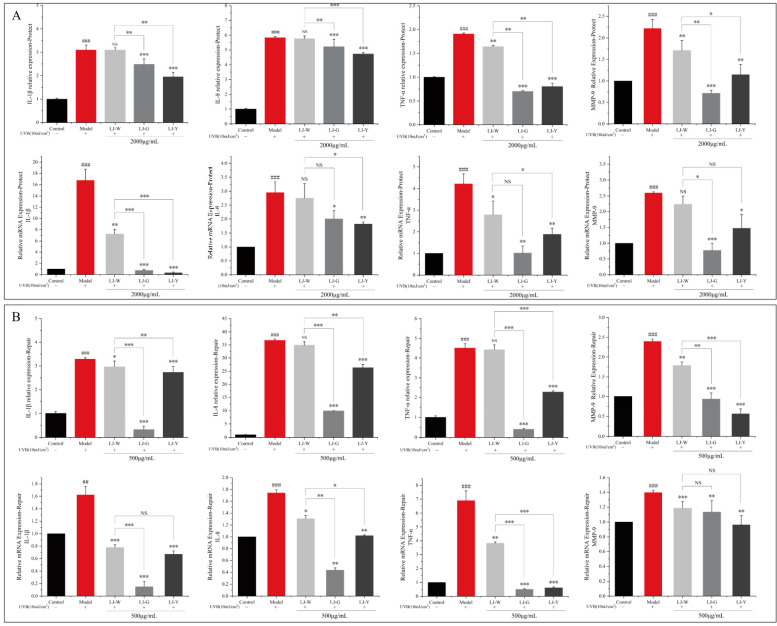
Effects of LJ-W, LJ-G and LJ-Y on the secretion of inflammatory factors in UVB-damaged HaCaT cells. (**A**): Differences in the content of IL-1β, IL-8, TNF-α and MMP-9 in HaCaT cells in UVB-damaged protection group; (**B**): Differences in the content of IL-1β, IL-8, TNF-α and MMP-9 in HaCaT cells in UVB-damaged repair group. * *p* < 0.05, ** *p* < 0.01, *** *p* < 0.001 compared with the model and LJ-W groups, ^NS^ *p* > 0.05, ^##^ *p* < 0.01, ^###^
*p* < 0.001 compared with the control group.

**Figure 6 marinedrugs-20-00650-f006:**
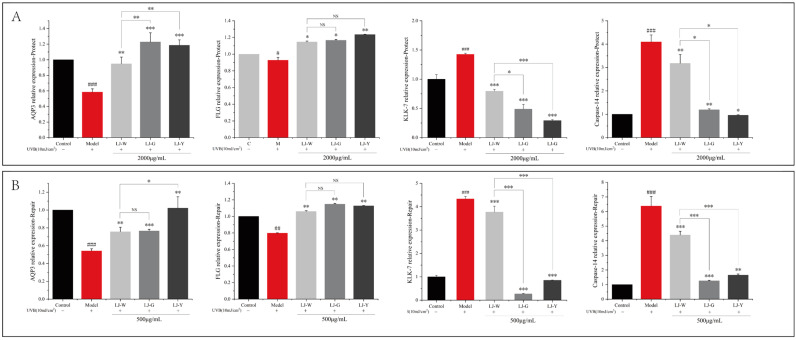
Effects of LJ-W, LJ-G and LJ-Y on the secretion of skin barrier factors in UVB-damaged HaCaT cells. (**A**): Differences in the content of AQP3, FLG, KLK-7 and Caspase-14 in HaCaT cells in UVB-damaged protection group; (**B**): Differences in the content of AQP3, FLG, KLK-7 and Caspase-14 in HaCaT cells in UVB-damaged repair group. * *p* < 0.05, ** *p* < 0.01, *** *p* < 0.001 compared with the model and LJ-W groups, ^NS^ *p* > 0.05, ^#^ *p* < 0.05, ^##^ *p* < 0.01, ^###^
*p* < 0.001 compared with the control group.

**Figure 7 marinedrugs-20-00650-f007:**
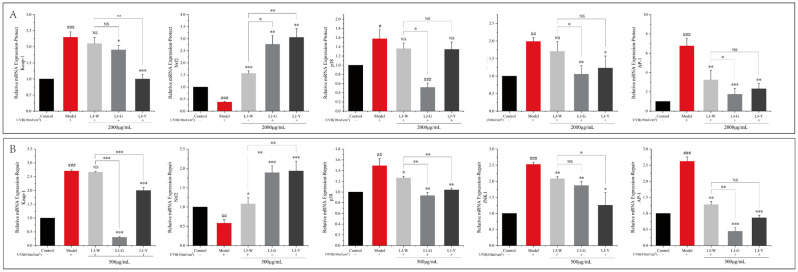
Effects of LJ-W, LJ-G and LJ-Y on Keap-1, Nrf2, p38, JNK1 and AP-1 genes in the Nrf2 signaling pathway. (**A**): UVB-damaged protection group; (**B**): UVB-damaged repair group. * *p* < 0.05, ** *p* < 0.01, *** *p* < 0.001, compared with the model and LJ-W groups, ^NS^ *p* > 0.05, ^#^ *p* < 0.05, ^##^ *p* < 0.01, ^###^ *p* < 0.001 compared with the control group.

**Figure 8 marinedrugs-20-00650-f008:**
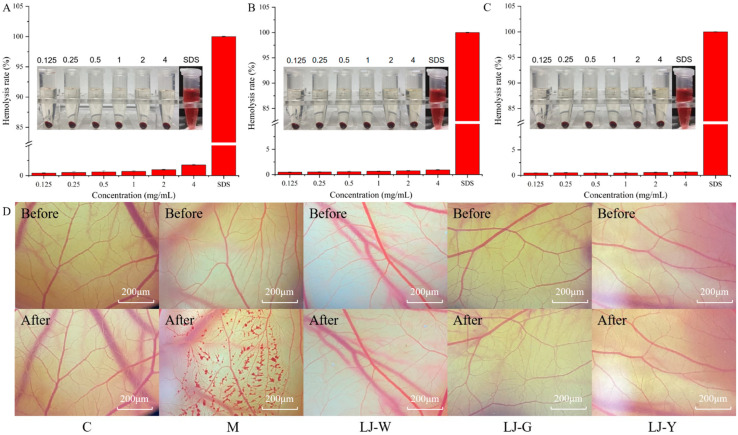
LJ-W, LJ-G and LJ-Y safety evaluation. (**A**): Effects of LJ-W on the hemolysis of rabbit erythrocytes; (**B**): Effects of LJ-G on the hemolysis degree of rabbit erythrocytes; (**C**): Effects of LJ-Y on the hemolysis degree of rabbit erythrocytes; (**D**): Effects of LJ-W, LJ-G and LJ-Y on the vascular damage of chick embryo chorioallantoic membrane.

**Figure 9 marinedrugs-20-00650-f009:**
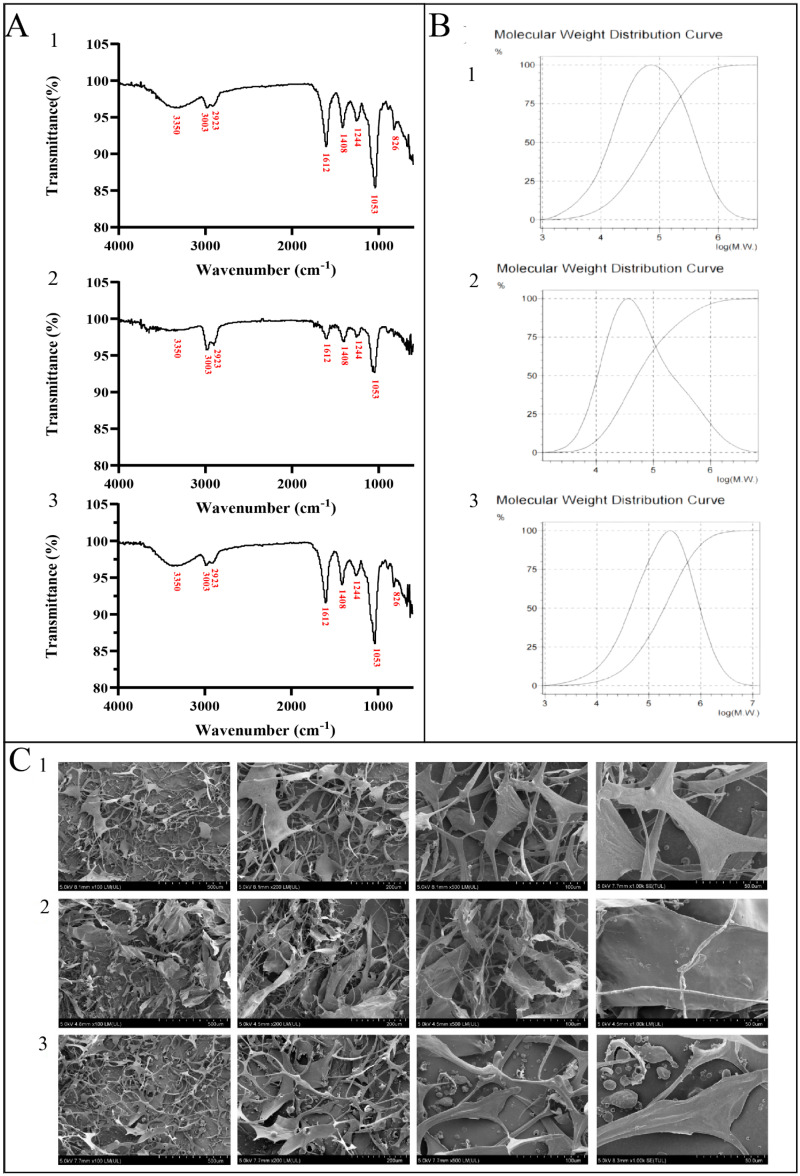
Morphological and structural characterization of unfermented and fermented *Laminaria japonica*. (**A**): IR spectrum of LJ-Wp (1), LJ-Gp (2) and LJ-Yp (3); (**B**): GPC Chromatogram of LJ-Wp (1), LJ-Gp (2) and LJ-Yp (3); (**C**): SEM image of LJ-Wp (1), LJ-Gp (2) and LJ-Yp (3) in 500, 200, 100 and 50 nm.

**Figure 10 marinedrugs-20-00650-f010:**
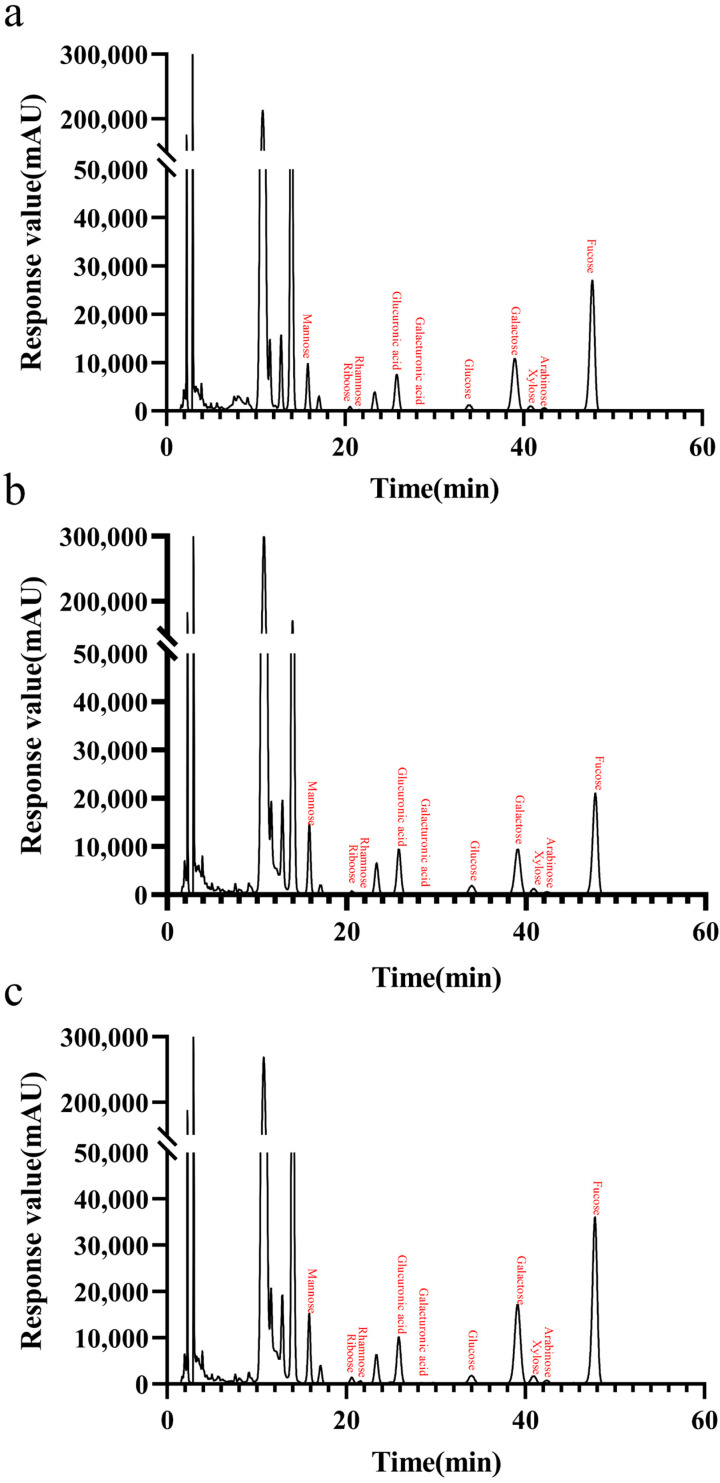
Monosaccharide components of LJ-Wp (**a**), LJ-Gp (**b**) and LJ-Yp (**c**). Detection wavelength is 250 nm, and detection interval is 500 ms.

**Table 1 marinedrugs-20-00650-t001:** Contents of total and reduction sugars and phenolic compounds analyzed by spectrophotometric methods before and after fermentation by *Laminaria japonica*. Data is represented as mean ± SEM. *** *p* < 0.001, significantly different compared to the LJ-W group.

Sample	Total Sugars (mg/g)	Polysaccharide (mg/g)	Total Phenols (mg/g)	Protein (mg/g)
LJ-W	34.56 ± 0.95	32.27 ± 2.95	0.47 ± 0.10	1.39 ± 0.19
LJ-G	81.25 ± 0.85 ***	78.99 ± 6.98 ***	0.86 ± 0.15 ***	4.66 ± 0.29 ***
LJ-Y	68.97 ± 5.19 ***	67.08 ± 5.71 ***	0.88 ± 0.06 ***	3.45 ± 0.5 ***

**Table 2 marinedrugs-20-00650-t002:** Classification of types of hemorrhagic effects of *Laminaria japonica* water extract and fermentation broths on CAM.

Sample	Type of Bleeding	Rating/Point						ES/Point
		1	2	3	4	5	6	
Negative control (0.9% NaCl solution)	Bleeding	0	0	0	0	0	0	0
	Blood vessel coagulation	0	0	0	0	0	0	
	Angiolysis	0	0	0	0	0	0	
Positive control (0.1 mol/L NaOH)	Bleeding	2	2	2	2	2	2	12
	Blood vessel coagulation	2	1	1	1	2	1	
	Angiolysis	1	1	1	1	1	1	
LJ-W	Bleeding	0	0	0	0	0	0	0
	Blood vessel coagulation	0	0	0	0	0	0	
	Angiolysis	0	0	0	0	0	0	
LJ-G	Bleeding	0	0	0	0	0	0	0
	Blood vessel coagulation	0	0	0	0	0	0	
	Angiolysis	0	0	0	0	0	0	
LJ-Y	Bleeding	0	0	0	0	0	0	0
	Blood vessel coagulation	0	0	0	0	0	0	
	Angiolysis	0	0	0	0	0	0	

**Table 3 marinedrugs-20-00650-t003:** LJ-Wp, LJ-Gp and LJ-Yp molecular weight determinations.

Name	LJ-Wp	LJ-Gp	LJ-Yp
Mn (Da)	2.9466 × 10^4^	2.8208 × 10^4^	6.2539 × 10^4^
Mw (Da)	16.1232 × 10^4^	18.1159 × 10^4^	39.6117 × 10^4^
Mz (Da)	54.8664 × 10^4^	102.6565 × 10^4^	302.8553 × 10^4^
Mw/Mn	5.47178	6.42222	6.33390
Mz/Mn	3.40296	5.66667	3.36240

**Table 4 marinedrugs-20-00650-t004:** Monosaccharide component of LJ-Wp, LJ-Gp and LJ-Yp.

Monosaccharide Ratio (%)	Sample		
	LJ-Wp	LJ-Gp	LJ-Yp
Mannose	11.789	13.903	9.696
Riboose	1.198	0.903	1.200
Rhamnose	0.749	0.541	0.583
Glucuronic acid	10.272	13.940	10.044
Galacturonic acid	0.422	0.533	0.371
Glucose	3.230	4.815	3.114
Galactose	22.094	21.020	26.090
Xylose	2.722	2.959	2.808
Arabinose	2.057	1.766	1.419
Fucose	45.467	39.620	44.675
Total	100	100	100

**Table 5 marinedrugs-20-00650-t005:** Reverse transcription system.

Reagent Name	Volume (μL)
Total RNA	2.0
Anchored Oligo(Dt)18 Primer	1.0
2 × ES Reaction Mix	10.0
EasyScript RT/RI Enzyme Mix	1.0
Gdna Remover	1.0
Rnase-free Water	5.0

**Table 6 marinedrugs-20-00650-t006:** Primer sequences for Real-Time PCR.

Gene	Direction	Primer Pair Sequence (5′→3′)
HO-1	F R	CAAGCGCTATGTTCAGCGAC GCTTGAACTTGGTGGCACTG
IL-1β	F R	CCTGAGCTCGCCAGTGAAA GTGGTGGTCGGAGATTCGTA
TNF-α	F R	CACAGTGAAGTGCTGGCAAC AGGAAGGCCTAAGGTCCACT
Keap-1	F R	GGAGGCGGAGCCCGA GATGCCCTCAATGGACACCA
AP-1	F R	TCTCAACATGGGTGGTCTGT AAATGCTTCATGCGGCGAAG
IL-8	F R	AAGATGTGAAGCTGACGCAGA AGAATTGAGCTGAGCCTTGG
MMP-9	F R	GTACTCGACCTGTACCAGCG AGAAGCCCCACTTCTTGTCG
JNK1	F R	CTGTGTGGAATCAAGCACCTTCA CTGGCCAGACCGAAGTCAAGA
p38	F R	TTAACAGGATGCCAAGCCATGA GGCACCAATAAATACATTCGCAAAG
Nrf2	F R	CAACTCAGCACCTTGTATC TTCTTAGTATCTGGCTTCTT
NQO1	F R	CAGCCAATCAGCGTTCGGTA CTTCATGGCGTAGTTGAATGATGTC

F: forward primer; R: reverse primer.

**Table 7 marinedrugs-20-00650-t007:** Reagents and dosage.

Reagent Name	Volume (μL)
Template	1.5
Forward Primer (10 μM)	0.4
Reverse Primer (10 μM)	0.4
2 × TransStart^®^ Top Green qPCR SuperMix	10.0
Passive Reference Dye (50×)	0.4
Nuclease-free Water	7.3

## Data Availability

Such information is available from the corresponding author upon reasonable request.

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
