# Peer review of "Two *Laminaria japonica* Fermentation Broths Alleviate Oxidative Stress and Inflammatory Response Caused by UVB Damage: Photoprotective and Reparative Effects"

_marinedrugs, 2022, doi:10.3390/md20100650_

Round 1

Reviewer 1 Report

This is a good article. The purpose of the work is well stated. The methods are adequate and the results clearly show the differences between the different treatments, so I think the article can be published in its current state.

Author Response

Dear Reviewers,
Thank you for your great recognition and approval of our articles. Your encouragement gives us more confidence in our future research path.

Reviewer 2 Report

Dear Authors, here are my comments:

The introduction is nicely written and divided into sections so that the reader can gradually get into the subject. My suggestion is that you might shorten the introduction a bit, as it is quite lengthy.
Results: It would be good if you placed the table and images in the results after the introductory sentences rather than directly after the chapter heading.
Section 2.4. figure 3 A. the picture is too small, you can almost not see anything, I recommend you to upload a bigger picture.
Figures 4, 5 and 6. it would be good if the graphs were bigger and more visible to the reader, maybe if you put one under the other.
The results are clearly presented, as is the discussion, although the discussion lacks comparisons with other authors' work.

Reviewer 3 Report

This manuscript demonstrated the effect of Laminaria japonica on UVB-damaged epidermal inflammation. This finding is interesting and informative. However, some points should be addressed as followings before acceptable.

1.              It would be better to increase the data quality of figures (especially fig. 1~7)

2.              Conclusion is missing.

3.              A reference for 426-427 line is missing.

4.              The author should compare JJ-W, JG-G, and JJ-Y, but did compare only to the LJ-W group in some results or to the control group.
